

# Evolution history of duplicated *smad3* genes in teleost: insights from Japanese flounder, *Paralichthys olivaceus*

Xinxin Du, Yuezhong Liu, Jinxiang Liu, Quanqi Zhang and Xubo Wang

Key Laboratory of Marine Genetics and Breeding, Ministry of Education, College of Marine Life Sciences, Ocean University of China, Qingdao, Shandong, China

## ABSTRACT

Following the two rounds of whole-genome duplication (WGD) during deuterosome evolution, a third genome duplication occurred in the ray-fined fish lineage and is considered to be responsible for the teleost-specific lineage diversification and regulation mechanisms. As a receptor-regulated SMAD (R-SMAD), the function of *SMAD3* was widely studied in mammals. However, limited information of its role or putative paralogs is available in ray-finned fishes. In this study, two *SMAD3* paralogs were first identified in the transcriptome and genome of Japanese flounder (*Paralichthys olivaceus*). We also explored *SMAD3* duplication in other selected species. Following identification, genomic structure, phylogenetic reconstruction, and synteny analyses performed by MrBayes and online bioinformatic tools confirmed that *smad3a/3b* most likely originated from the teleost-specific WGD. Additionally, selection pressure analysis and expression pattern of the two genes performed by PAML and quantitative real-time PCR (qRT-PCR) revealed evidence of subfunctionalization of the two *SMAD3* paralogs in teleost. Our results indicate that two *SMAD3* genes originate from teleost-specific WGD, remain transcriptionally active, and may have likely undergone subfunctionalization. This study provides novel insights to the evolution fates of *smad3a/3b* and draws attentions to future function analysis of *SMAD3* gene family.

Corresponding author
Xubo Wang, wangxubo@ouc.edu.cn

## INTRODUCTION

SMAD transcription factors are considered as the core of the TGF-$\beta$ pathway, which are activated by membrane receptors and regulate target genes by transcriptional complexes (*Massagué, Seoane & Wotton, 2005*). According to the functional variety, SMADs can be divided into three subfamilies: receptor-activated SMADs (R-SMADs: *SMAD1, SMAD2, SMAD3, SMAD5, and SMAD8*), common mediator SMADs (Co-SMADs: *SMAD4*), and inhibitory SMADs (I-SMADs: *SMAD6, SMAD7*) (*Miyazono, Ten Dijke & Heldin, 2000*; *Moustakas, Souchelnytskyi & Heldin, 2001*). SMADs contain two conserved structural domains—the N-terminal MH1 domain and the C-terminal MH2 domain—and a linker region with multiple phosphorylation sites between the two domains (*Shi & Massagué, 2003*). As an R-SMAD, *SMAD3* has various functions including regulating the

pathogenesis of diseases and even cancer progression (*Bonniaud et al., 2004*; *Ge et al., 2011*; *Roberts et al., 2006*).

To date, *SMAD* genes have been found only in eumetazoan animals. Four *SMAD* genes have been identified in fruit fly *Drosophila melanogaster*, seven in nematode *Caenorhabditis elegans* and eight in human *Homo sapiens* (*Newfeld & Wisotzkey, 2006*). Nevertheless, much more novel *SMADs* exist in teleost, such as *SMAD2, SMAD3,* and *SMAD6.* As described in previous studies, these novel isoforms are highly similar and evidently caused by an additional round of whole-genome duplication (WGD) in teleost fishes rather than fragment duplication or alternative splicing (*Huminiecki et al., 2009*; *Pang et al., 2011*; *Sato & Nishida, 2010*).

The increased complexity and genome size of vertebrates have been derived from two verified rounds of WGD, which are thought to play major roles in promoting diversification and evolutionary innovation within vertebrates (*Cañestro et al., 2013*; *Dehal & Boore, 2005*; *Hoegg & Meyer, 2005*; *Hoffmann, Opazo & Storz, 2011*). Moreover, a fish-specific genome duplication (FSGD or 3R) occurred ~350 million years ago which resulted in new copies of genes and provided genetic basis for evolutionary innovation (*Meyer & Schartl, 1999*; *Meyer & Van de Peer, 2005*; *Taylor et al., 2001*; *Vandepoele et al., 2004*). According to the study of an additional lineage-specific WGD in salmonids and some cyprinids, the climatic cooling and subsequent evolution of anadromy are major catalyst for salmonid speciation rather than the WGD itself (*Berthelot et al., 2014*; *Glasauer & Neuhauss, 2014*; *Macqueen & Johnston, 2014*). It is indicated that 3R is not directly associated with species diversity but 3R-derived duplicate genes may have subsequently undergone dosage effects regulation, lineage-specific evolution and been divergent in regulatory mechanisms, expression pattern and evolutionary rates after lineage diversification (*Braasch, Salzburger & Meyer, 2006*; *Mulley, Chiu & Holland, 2006*; *Sato & Nishida, 2010*; *Siegel et al., 2007*). According to the duplication-degeneration-complementation (DDC) model, duplicated genes will undergo three main fates, namely, nonfunctionalization (duplicates dying out as pseudogenes), subfunctionalization (partitioning of ancestral gene functions on the duplicates) and neofunctionalization (assigning a novel function to one of the duplicates) (*Force et al., 1999*).

In teleost zebrafish, it has been reported that *SMAD3* is required for regenerative capacity of heart and mesendoderm induction (*Chablais & Jaźwińska, 2012*; *Jia et al., 2008*). However, researches investigating the origination and evolution fates of teleost *SMAD3* paralogs are deficient. To better understand the origin and functional diversification of *SMAD3* paralogs in teleost, we identified whole set of *SMAD* gene family sequences from the transcriptome and genome of Japanese flounder *Paralichthys olivaceus* and other teleosts. Next, gene structure, phylogenetic reconstruction, and chromosomal synteny analyses of vertebrate *SMAD3* genes were performed to study the origin and evolution of two *SMAD3* genes in teleosts. The analyses of motif scan, positive selection, and expression profiles of the two *SMAD3* genes in Japanese flounder were performed to identify potential functional changes for the duplicated *SMAD3* genes within the lineage of teleost. The results provide evidences to the duplication of teleost *SMAD* gene family derived from the WGD and possible subfunctionalization of teleost-specific duplicated

*SMAD3* genes. Moreover, this study lays the foundation for evolutionary and functionary studies of *SMAD3* gene family in teleosts.

## MATERIALS AND METHODS

### Ethics statement

Japanese flounder samples were collected from local aquatic farms. This research was conducted in accordance with the Institutional Animal Care and Use Committee of the Ocean University of China and the China Government Principles for the Utilization and Care of Vertebrate Animals Used in Testing, Research, and Training (State science and technology commission of the People's Republic of China for No. 2, October 31, 1988. http://www.gov.cn/gongbao/content/2011/content_1860757.htm).

### Fish

Healthy two-year-old Japanese flounder (three females and three males) were selected from a larger cohort population. The flounders were anesthetized and killed by severing spinal cord. Organs, including heart, liver, spleen, kidney, brain, gill, muscle, intestine, and gonad, were collected in triplicate from each fish. Samples were immediately frozen by liquid nitrogen and stored at −80 °C for extraction of total RNA.

### Identification of *SMAD* genes in Japanese flounder and other species

The *SMAD* coding sequences of Amazon molly (*Poecilia formosa*), Japanese medaka (*Oryzias latipes*), and Nile tilapia (*Oreochromis niloticus*) were used as queries for local TBLASTX searches with an E-value of 1e-5 against the genome (Q. Zhang, 2016, unpublished data) and transcriptome (SRA, accession number: SRX500343) of Japanese flounder to identify the DNA sequences of *SMAD*. The coding sequences of other species (green spotted puffer *Tetraodon nigroviridis*, tongue sole *Cynoglossus semilaevis*, bicolor damselfish *Stegastes partitus*, gubby *Poecilia reticulate*, platyfish *Xiphophorus maculatus*) were obtained from NCBI or Ensembl database, and the accession numbers were shown in Table S1. Different abbreviations for *SMAD* gene orthologs exist in NCBI. For convenience reasons, *SMAD* was used for all vertebrate orthologs and *smad3a* and *smad3b* for the variants identified in teleost in this study.

### Phylogenetic analysis of *SMAD* genes

In order to study the phylogenetic relations and evolution fates of *SMAD* genes, a phylogenetic reconstruction including all *SMAD* isoforms of vertebrates was performed. The whole coding sequences of *SMAD* isoforms were aligned by Clustal X with the default parameters (*Chenna et al., 2003*). Sequences used to construct phylogenetic trees were retrieved from NCBI and Ensembl (species names, gene names and accession numbers are available in Table S1). Appropriate substitution model of molecular evolution, GTR+I+G, was determined by JModelTest v2.1.4 (*Darriba et al., 2012*). Phylogenetic tree was constructed by Bayesian method which was implemented in MrBayes v3.2.2 (*Huelsenbeck & Ronquist, 2001*; *Ronquist et al., 2012*).

A second phylogenetic reconstruction including only *SMAD3* isoforms was performed to confirm phylogenetic relations between *smad3a* and *smad3b*. The whole coding sequences of all *SMAD3* isoforms were aligned by Clustal X with default parameters. Phylogenetic trees were constructed by Bayesian method and maximum likelihood method with GTR+I+G substitution model, respectively. Maximum likelihood phylogeny was constructed by phyML v3.1, and the branching reliability was tested by bootstrap resampling with 1,000 replicates (*Guindon et al., 2010*).

## Genomic structure, motif, and synteny analysis of teleost *SMAD3* paralogs

The exon-intron information of teleost *SMAD3* genes was obtained by BLASTn with coding sequences against the corresponding genomic sequences. Figures of teleost *SMAD3* genomic structures were obtained using an online tool Gene Structure Display Server 2.0 (GSDS: http://gsds.cbi.pku.edu.cn) with size and position information of each exon and intron (*Hu et al., 2015*). Alignments of the teleost *smad3a/3b* amino acids sequences were constructed by Clustal X (*Chenna et al., 2003*). MEME was applied to identify motifs of the *SMAD3* coding sequences to test the possible functional divergence between teleost *smad3a* and *smad3b* (*Bailey et al., 2009*). Synteny comparisons of the fragments harboring *SMAD3* and flanking genes were performed to test the genes' syntenic conservation. Flanking genes of *smad3a/3b* used in the synteny analysis were extracted from online genome databases, such as Ensembl or NCBI. The genes were mapped according to their relative locations in the chromosome for the synteny analysis. In order to test the chromosomal synteny conservation between human and green spotted puffer, an online synteny database was used to analyze the syntenic conservation between human chromosome Hsa15, where human *SMAD3* gene was located, and genome of green spotted puffer (*Catchen, Conery & Postlethwait, 2009*).

## Positive selection test of teleost *smad3a* and *smad3b* genes

As described in the manual of PAML v4.7, the species used in the analysis should have a close genetic relationship, absolute minimum species used in the analysis is four or five, and the dS summed over all branches on the tree should be larger than 0.5. In accordance with the criteria, nine teleost species were selected to explore differences in selective pressure between *smad3a* and *smad3b*. Phylogenetic tree used for positive selection analysis was constructed by MrBayes with GTR+I+G model. Various site models (M0, M1a, M2a, M3, M7, M8, and M8a) in CODEML were applied to estimate the ratio of nonsynonymous to synonymous substitutions (dN/dS = ω) and likelihood ratio tests (LRTs) to confirm the sites that were under positive selection (*Yang, 2007*). In the nested site models, M0 and M3 were compared to detect whether the dN/dS was changing. Moreover, the comparisons of M2a/M1a, M8/M7, and M8a/M8 were used to estimate the positively selected sites.

## RNA extraction, cDNA synthesis and distribution pattern of *SMAD3* genes in Japanese flounder

Total RNA was extracted from organ samples with Trizol reagent, according to the manufacturer' protocol (Invitrogen, Carlsbad, CA, USA). Then, DNase I (TaKaRa, Dalian, China) was applied to remove genomic DNA; protein was removed by RNAclean RNA kit (Biomed, Beijing, China). Agarose gel electrophoresis and NanoPhotometer Pearl (Implen GmbH, Munich, Germany) were used to evaluate the quality and quantity of RNA. cDNA was synthesized by M-MLV kit (TaKaRa) in accordance with the manufacturer's instructions.

Two specific primer pairs (Table S2) for Japanese flounder *smad3a* and *smad3b* were designed by an online tool IDT (http://www.idtdna.com/Primerquest/Home/Index), in the untranslated region of both genes. Each sample was pooled from three male or female individuals, which was performed in triplicate. Pre-experiment was conducted to test product specificity. quantitative real-time PCR (qRT-PCR) was performed with SYBR Premix Ex Taq II (TaKaRa) by LightCycler 480 (Roche, Forrentrasse, Switzerland) with thermocycling consisted of 5 min at 94 °C for pre-incubation, followed by 40 cycles at 94 °C (15 s) and 60 °C (45 s). As described, 18S rRNA was used as the reference gene to determine the relative expression (*Zhong et al., 2008*). The target gene's expression was analyzed by $2^{-\Delta\Delta Ct}$ method. qRT-PCR data were statistically analyzed by one-way ANOVA with SPSS 20.0. P < 0.05 was considered to indicate statistical significance. All data were expressed as mean ± standard error of the mean (SEM).

# RESULTS

## Identification of Japanese flounder *SMAD* gene family

To study the evolution history of *SMAD* gene family, a whole set of *SMAD* gene family (*SMAD1*, two *SMAD2*, two *SMAD3*, four *SMAD4*, *SMAD5*, two *SMAD6*, *SMAD7*, and *SMAD8*) was identified from Japanese flounder transcriptome and genome by TBLASTX with 1e-5. Integrated *SMAD* coding sequences in other specific species, such as human, house mouse, chicken, Nile tilapia, Japanese medaka, and zebrafish were also found from multiple databases. In this gene family, *SMAD2*, *SMAD3*, and *SMAD6* duplicates were identified only in teleosts, except for spotted gar, while other species had only single copy. It could be speculated that these novel *SMAD* isoforms might derive from a teleost-specific WGD.

## Genomic structures of teleost *SMAD3*

The differences between teleost *smad3a* and *smad3b* genes were shown in gene structure graphic. *smad3b* gene had one more exon but shorter gene full-length than *smad3a* because of one long intron in *smad3a*. The lengths of each corresponding exon were highly conserved between these two genes. Besides, the fourth exon in *smad3a* was divided into two exons in *smad3b* by an additional intron resulting in the different exon numbers (Fig. S1). Multiple sequence alignment of deduced full-length SMAD3a and SMAD3b protein sequence was constructed with selected teleost species (i.e., Japanese medaka, Japanese flounder, Amazon molly, and tongue sole) to test the similarity between the

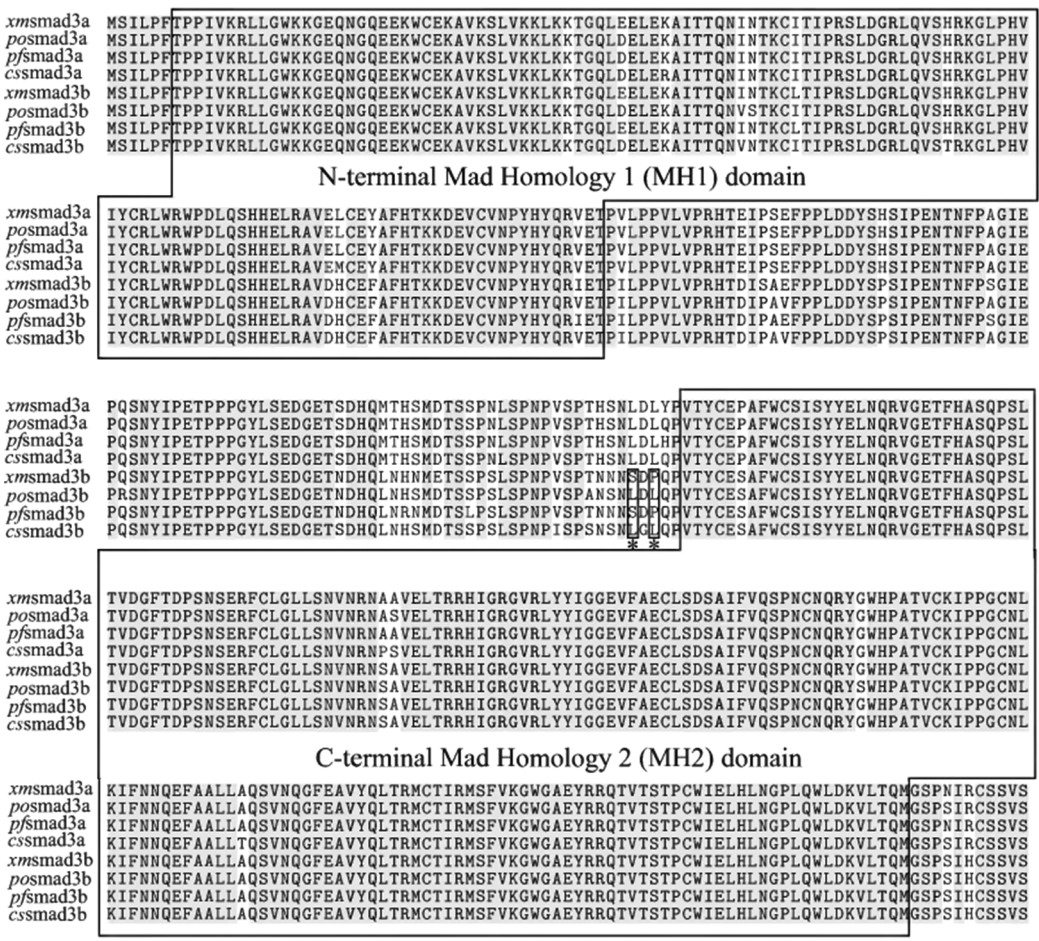

**Figure 1 Sequence alignment of the deduced SMAD3a and SMAD3b protein sequences.** Identical amino acids are in gray background. Two conserved domains, namely, MH1 and MH2 domains are marked in the figure. Two significantly positively selected sites are indicated by asterisk.

paralogs. The sequences of SMAD3a and SMAD3b were highly similar and Mad homology domains were conserved in both N-terminal and C-terminal of SMAD3a/3b. The specific mutations at MH1 and MH2 domains between the two paralogs such as tyrosine to phenylalanine, proline to serine, and in linker region such as serine to alanine, histidine to proline might lead to functional disparities between the two isoforms (Fig. 1). Taken together, it could be implied that the differences between the two *SMAD3* isoforms might lead to a functional diversification.

## Phylogenetic reconstruction of *SMAD*

A phylogenetic tree constructed by Bayesian method with GTR+I+G model of all vertebrate *SMAD* isoforms was applied to test the evolution fates of *SMAD* gene family. Results indicated the relationships of *SMAD* gene family members. *SMAD* genes distinctly divided into four subfamilies, namely, *SMAD1/5/8*, *SMAD2/3*, *SMAD4*, and *SMAD6/7*. In addition, four fruit fly genes (*Mad*, *Smox*, *Medea*, and *Dad*) were clustered to homologous subfamily. Teleost-specific *SMAD2*, *SMAD3*, and *SMAD6* paralogs were

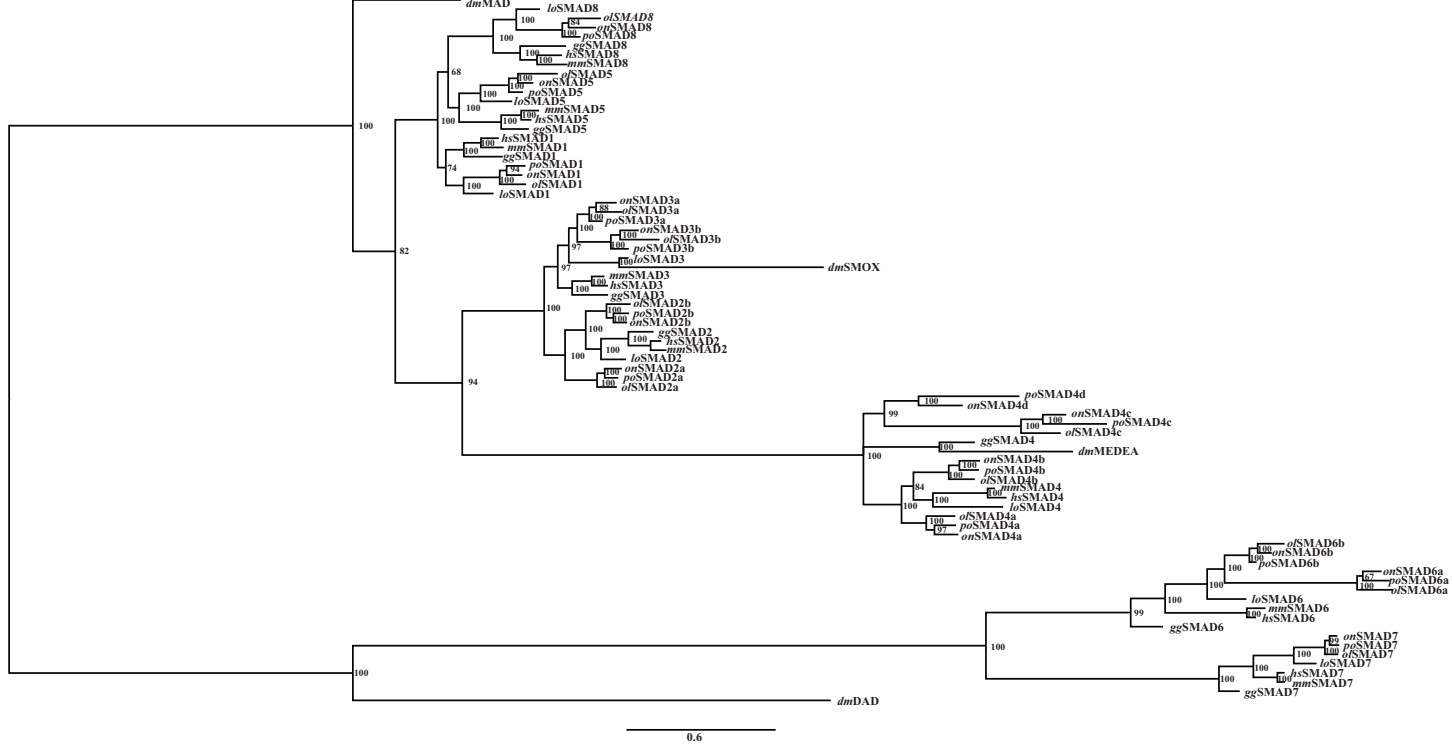

**Figure 2 Phylogenetic analyses of *SMAD* gene family.** Phylogenetic tree constructed by Bayesian method with GTR+I+G, MCMC = 800,000. Numbers at the tree nodes are posterior probabilities. *Po, Paralichthys olivaceus; Hs, Homo sapiens; Mm, Mus musculus; Gg, Gallus gallus; On, Oreochromis niloticus; Lo, Lepisosteus oculatus; Dm, Drosophila melanogaster; Ol, Oryzias latipes.*

clustered into one clade and then clustered with other orthologs implying that *SMAD* gene family had duplicated and were most likely resulted from the WGD (Fig. 2).

To study the duplication of *SMAD3* in the lineage of teleost, a second phylogenetic reconstruction including only *SMAD3* isoforms was performed by Bayesian and maximum likelihood method, respectively with GTR+I+G model and the elephant shark *SMAD3* sequence was set as an outgroup. Similar topologies were inferred by the two programs (Fig. 3). Results indicated that these species gathered into two main clades, i.e., teleost clade and non-teleost clade. Teleost *SMAD3* genes could be clearly divided into two well-conserved clusters, i.e., *smad3a* and *smad3b*, whereas spotted gar *SMAD3* occupied a separate clade. Moreover, the branch length of *smad3b* cluster was longer than *smad3a*, and the branch lengths of frog *SMAD3* and anole lizard *SMAD3* were also longer than those in the other classes. These results implied that duplication of *SMAD3* was widespread in teleost except for spotted gar, a species never experienced the teleost-specific WGD (*Braasch et al., 2016*). Combining these two phylogenetic trees, it could be deduced that teleost *SMAD3* paralogs might originate from the teleost-specific WGD.

## Synteny analysis of *SMAD3* genes

Synteny analyses were applied to testify the speculation that teleost *SMAD3* paralogs originated from teleost-specific WGD rather than fragment duplication or alternative splicing. As shown in Fig. 4, the *SMAD3* genes and adjoining genes were placed

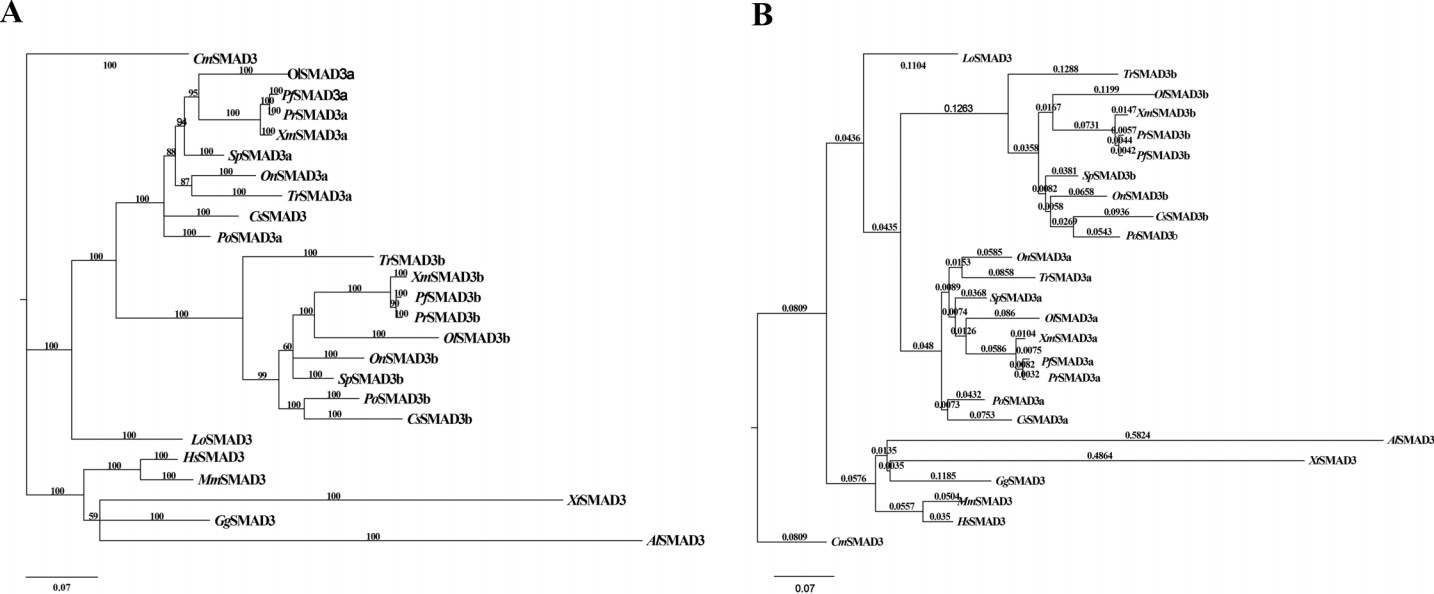

**Figure 3 Phylogenetic analyses of *SMAD3*.** (A) Phylogenetic tree constructed by Bayesian method with GTR+I+G, MCMC = 300,000. Elephant shark *SMAD3* was used as the outgroup. Numbers at the nodes are posterior probabilities. (B) Phylogenetic tree constructed by phyML with GTR+I+G. Elephant shark *SMAD3* was used as the outgroup. Numbers at the nodes are bootstrap values with 1,000 replicates. *Cm, Callorhinchus milii; Lo, Lepisosteus oculatus; Tr, Takifugu rubripes; Ol, Oryzias latipes; Xm, Xiphophorus maculatus; Pr, Poecilia reticulata; Pf, Poecilia formosa; Sp, Stegastes partitus; On, Oreochromis niloticus; Cs, Cynoglossus semilaevis; Po, Paralichthys olivaceus; Al, Anolis carolinensis; Xt, Xenopus tropicalis; Gg, Gallus; Mm, Mus musculus; Hs, Homo sapiens.*

according to their relative locations on the scaffold or chromosome. The genes near teleost *smad3a* were highly conserved in teleost and shared the same direction. Similar results could be obtained from synteny analysis of *smad3b*. Comparison between *smad3a* and *smad3b* revealed that only *SMAD3*, *SMAD6*, and other five upstream genes (*kif23, tle3, morf41l, pias1,* and *skor1*) were conserved, and no paralogous genes were retained in the downstream of the two *SMAD3* paralogs. Long fragments consisting of several genes were lost in the upstream and downstream regions of *smad3a* in Japanese medaka and platyfish.

After comparing teleost *smad3a* and *smad3b* neighborhood gene sequences with other species, such as elephant shark, spotted gar, frog, anole lizard, chicken, and eucherian species, we found that adjoining genes of teleost *smad3b* shared highly conserved synteny with these species (Fig. S2). However, conserved synteny between teleost *smad3a* and *SMAD3* sequences in these species existed only in the upstream region. Thus, the teleost *smad3b* gene was more likely to be the ancestor *SMAD3* gene and *smad3a* derived from genome duplication. Moreover, long fragments were lost in the upstream regions of *smad3b* in spotted gar, elephant shark and anole lizard, and an inversion existed in the upstream region of teleost and other species such as chicken and ectherian. To some extent, the synteny results suggested that two teleost *SMAD3* paralogs originated from WGD.

To further verify the speculation, a chromosomal synteny test was performed between the genome of human and green spotted puffer using online synteny databases to

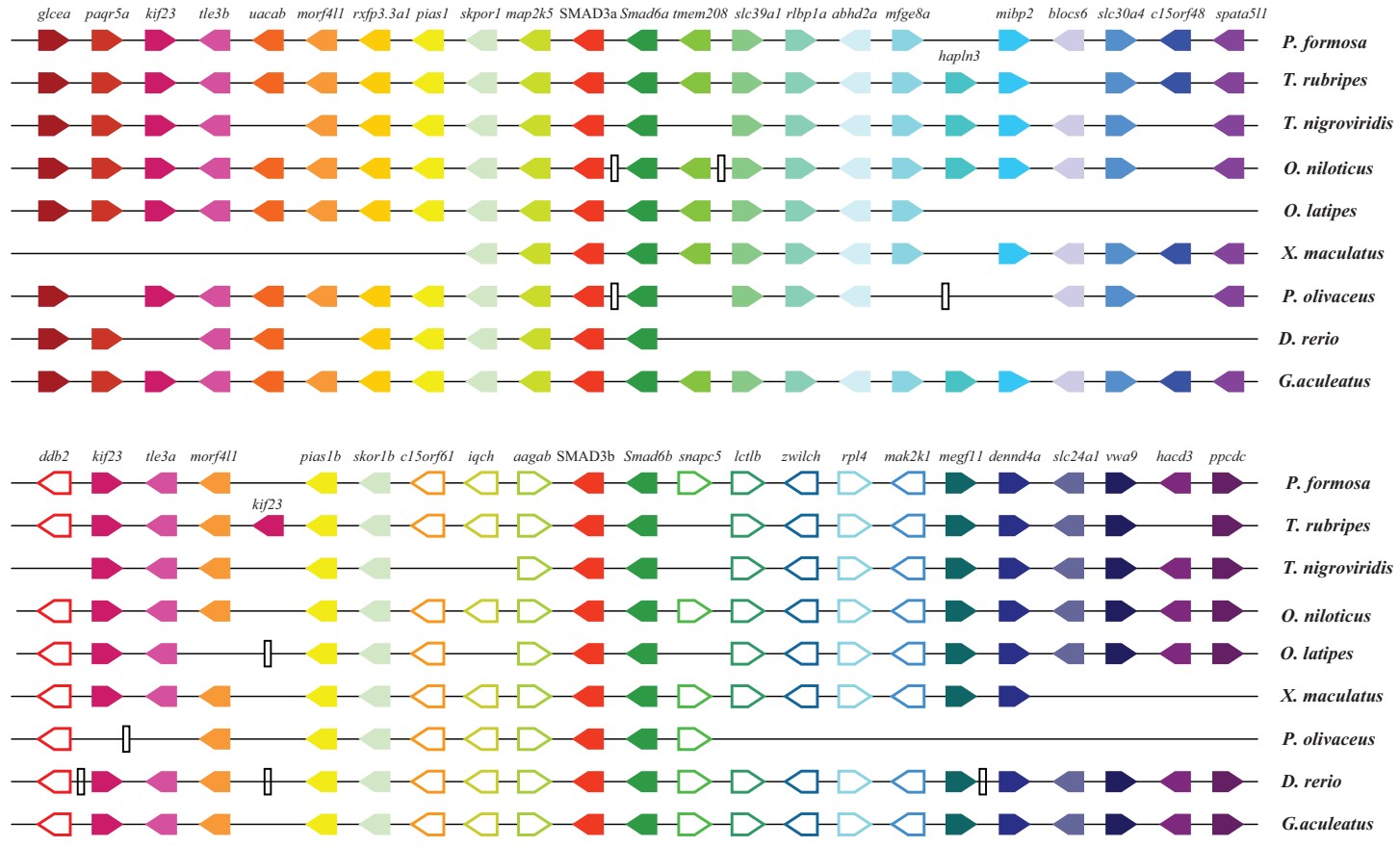

**Figure 4 Chromosomal segments showing the synteny of *smad3a* and *smad3b* in teleost.** Different genes are represented by different colored pentagons and gene order is determined according to their relative positions in the chromosome or scaffold; the gene names are placed on top of the pentagons. The direction of pentagons indicate the gene direction, the vertical lines represent noncontiguous regions on the scaffold or chromosome.

determine whether *SMAD3* paralogs originated from WGD. The human *SMAD3* gene was located at chromosome Hsa15, whereas green spotted puffer *smad3a* and *smad3b* were located at chromosomes Tni5 and Tni13, respectively. According to the chromosomal synteny dot plot (Fig. 5), the human *SMAD3* region showed double conserved synteny with green spotted puffer chromosomes Tni5 and Tni13. In addition, highly conserved synteny could also be detected between green spotted puffer chromosomes Tni5 and Tni13. Combining the gene neighborhood analysis and chromosomal synteny analysis, it could be speculated that these two *SMAD3* paralogs originated from a common ancestral gene during the teleost-specific WGD.

## Molecular evolution of teleost *smad3a* and *smad3b*

Multiple single nucleotide polymorphisms and random mutagenesis are found in protein sequence, and each substitution may have the potential to affect protein function (*Ng & Henikoff, 2003*). To test this potential in *SMAD3* genes, we examined the protein sequence evolution in *smad3a/3b* by codon-based models in PAML with three model pairs M0/M3, M1a/M2a, and M7/M8 (Table 1). The phylogenetic tree used for positive selection analysis is shown in Fig. S3.

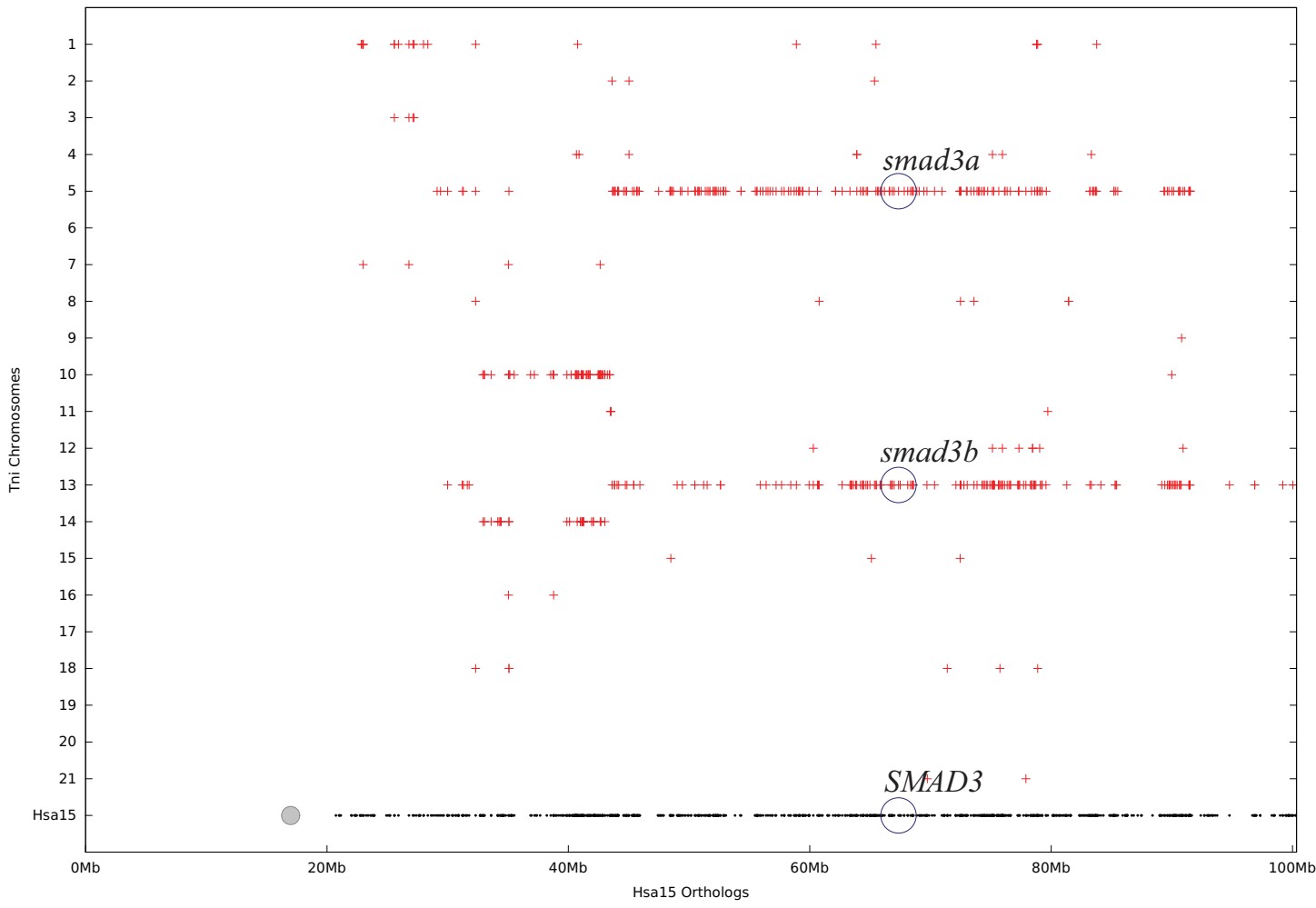

**Figure 5 Chromosome synteny of teleost *SMAD3* paralogs.** The dot plot between human *SMAD3* region and green spotted puffer genome indicates that human *SMAD3* gene region in chromosome Hsa15 shares double conservation with green spotted puffer *smad3a* gene region in chromosome Tni5 and *smad3b* gene region in chromosome Tni13. The black dots represent segments in human chromosome Hsa15, and the red dots represent the conserved segments in green spotted puffer genome which mostly located in chromosome Tni5 and Tni13.

According to the comparison between M3 and M0, it could be reflected that M3 was significantly better than M0 (P < 0.05) and M0 was rejected. Thus, both *smad3a* and *smad3b* were under variable alternative pressure. Comparison groups of M1a/M2a, M7/M8 and M8/M8a were then used to test the likelihood ratio. According to the results of chi2 program in PAML, we found that LRT significantly differed in M7/M8 pairs (P < 0.05) of *smad3b* and M8 showed better fitness to the data of *smad3b*, which was confirmed by an additional test between M8 and M8a. Bayes Empirical Bayes (BEB) method of M8 was applied to calculate the post probabilities of sites to identify the positive selected amino acid sites when LRT differed significantly. Five candidate positive selected sites were identified, two of which were significantly positively selected (219L**, 221L**, posterior probability > 0.99) in *smad3b*. However, no significantly positive selected sites were found in *smad3a* by comparisons (M1a/M2a, M7/M8 and M8/M8a). Positive selected sites were located in the linker region (Fig. 1). As a proline rich region, linker

**Table 1  Results of sites model analyses on the teleost *SMAD3* Bayesian gene tree.**

| Tree | Model | lnL | κ | Null | LRT | df | P-value | Site | BEB |
|---|---|---|---|---|---|---|---|---|---|
| SMAD3a | M0 | −4,084.415 | 3.41131 | NA | | | | | |
| | M1a | −4,066.241 | 3.11193 | NA | | | | | |
| | M2a | −4,066.241 | 3.11192 | M1a | 0 | 2 | 1.0000 | | |
| | M3 | −4,063.075 | 3.06893 | M0 | 42.68 | 4 | 0.0000 | | |
| | M7 | −4,067.745 | 3.02656 | NA | | | | | |
| | M8a | −4,065.670 | 3.08595 | NA | | | | | |
| | M8 | −4,065.563 | 3.08136 | M7 | 4.364 | 2 | 0.1128 | | |
| | | | | M8a | 0.214 | 1 | 0.6437 | | |
| SMAD3b | M0 | −4,822.425 | 2.24532 | NA | | | | | |
| | M1a | −4,735.378 | 2.37286 | NA | | | | | |
| | M2a | −4,734.233 | 2.34986 | M1a | 2.29 | 2 | 0.3182 | | |
| | M3 | −4,715.059 | 2.29131 | M0 | 214.732 | 4 | 0.0000 | | |
| | M7 | −4,734.999 | 2.32501 | NA | | | | | |
| | M8a | −4,718.436 | 2.30812 | NA | | | | | |
| | M8 | −4,714.852 | 2.29008 | M7 | 40.294 | 2 | **0.0000** | 219 (L) | 0.991** |
| | | | | | | | | 221 (L) | 0.994** |
| | | | | M8a | 7.168 | 1 | **0.0074** | | |

region determined the properties and functions of *SMAD* proteins by phosphorylation (*Kamato et al., 2013*). Mutations in this region may affect function of *smad3b* and result in functional diversification between *smad3a* and *smad3b*. These results indicated that there might be a functional diversification between teleost *smad3a* and *smad3b*.

### Expression pattern of Japanese flounder *smad3a* and *smad3b* genes

Different organ-specific expression patterns of Japanese flounder *smad3a* and *smad3b* could be deduced from the qRT-PCR results (Fig. 6). The *smad3a* and *smad3b* genes were expressed in all organs with different expression levels in specific organs. The expression levels of *smad3a* in spleen, kidney, and intestine were higher than *smad3b*, which were contrary to those in brain, gill, muscle, ovary, and testis. In addition, tissue distributions between *smad3a* and *smad3b* were significantly different in spleen, brain, muscle and ovary. The different expression patterns of Japanese flounder *smad3a* and *smad3b* might reflect the function divergence between the two genes.

## DISCUSSION

### Vertebrate *SMAD* gene family expanded by WGD

As a family of intracellular mediators, *SMAD* proteins are activated by serine/threonine kinase receptors and translocate signals from cytoplasm into nucleus to regulate the expression of target genes together with several kinds of transcription factors (*Attisano & Lee-Hoeflich, 2001*). With the development of next generation sequencing (NGS), we can easily identify *SMAD* gene sequences in genomes of related species. Eight *SMADs* were widely identified in vertebrate, namely, *SMAD1* to *SMAD8*, and divided into three

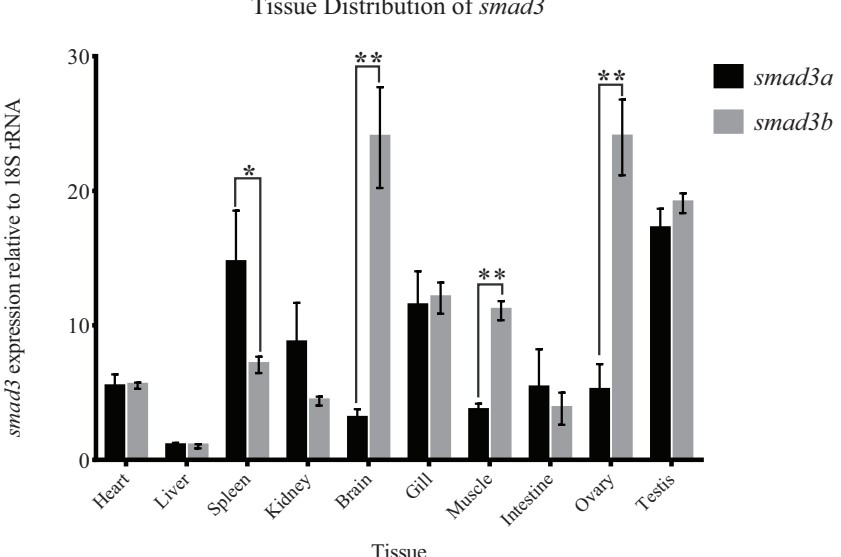

**Figure 6** **Expression patterns of *smad3a* and *smad3b* in Japanese flounder relative to 18S rRNA.** Data are shown as mean ± SEM (n = 3). Asterisks indicate statistical significance (P < 0.05).

subgroups by particular function (*Miyazono, Ten Dijke & Heldin, 2000*; *Moustakas, Souchelnytskyi & Heldin, 2001*). In addition, much more lineage-specific paralogs are identified. For example, frog has two *SMAD4s*, *SMAD4α*, and *SMAD4β* (*Masuyama et al., 1999*), and additional *SMAD2*, *SMAD3*, *SMAD6* isoforms exist in extant teleost fishes. It could be speculated that these novel isoforms are derived from WGD.

Phylogenetic analysis suggested that the *SMAD* gene family had undergone an expansion and was divided into *SMAD1/5/8*, *SMAD2/3*, *SMAD4*, and *SMAD6/7* subfamilies. As described in previous studies, the variation of *SMAD* gene family is a result of WGD (*Huminiecki et al., 2009*). 2R-WGD affected the majority of signaling genes with strongest effect on developmental pathways, such as receptor tyrosine kinases, Wnt, and TGF-$\beta$. The genes retained after 2R were enriched in protein interaction domains and multifunctional signaling modules of Ras and MAP-kinase cascades (*Huminiecki & Conant, 2012*). The phylogenetic analysis indicated that teleost-specific duplication occurred because more than one *SMAD2*, *SMAD3*, *SMAD4* and *SMAD6* paralogs existed in the teleost. In addition, different *SMAD* orthologs of each species shared conserved phylogenetic relationships. These findings suggested that *SMAD* gene family had undergone an expansion through WGD and resulted in many teleost-specific paralogs, such as *smad3a* and *smad3b*.

## Two teleost *SMAD3* isoforms are generated from 3R

New genes can arise from plenty of events, such as exon shuffling, gene duplication, retroposition, mobile elements, lateral gene transfer, gene fusion/fission and de novo origination (*Long et al., 2003*). Duplication is a pivotal means of generating genetic material for innovation. The two rounds of WGD have resulted in complex innovations in cellular networks. During the WGD period, many classes of genes such as transcription

factors, kinases, ribosomal protein, and cyclins, are duplicated more frequently (*Aury et al., 2006*; *Seoighe & Wolfe, 1999*; *Wolfe & Shields, 1997*). As a core member of the TGF-*β* family, *SMAD* has undergone duplication in 2R with an additional duplication in teleosts. Many paralogs have been degraded in the evolution history, which was consistent with dosage hypothesis (*Edger & Pires, 2009*; *Papp, Pál & Hurst, 2003*; *Qian & Zhang, 2008*), while additional *SMADs*, such as *SMAD2s*, *SMAD3s*, *SMAD4s*, and *SMAD6s*, were retained in teleost implying that teleost *smad3a* and *smad3b* are generated and retained from 3R.

According to the neighborhood gene synteny results, we could find that parts of adjoining genes were conserved around *smad3a/3b*, such as *smad6*, *skor1*, *pias*, *morf41*, *tle3*, and *kif23* (Fig. 4). The teleost *smad3b* flanking genes shared more conservation with chondrichthyes, amphibians, reptiles, birds, mammals and spotted gar which had only one *SMAD3* gene and did not undergo 3R (*Braasch et al., 2016*). The results support that *smad3a* and *smad3b* originate by WGD and that *smad3b* is more likely be the ancestral one.

To further verify the hypothesis, two *SMAD3s* genes were identified in green spotted puffer genome. Chromosomal synteny results demonstrated that chromosome Tni5 and Tni13 in green spotted puffer shared high conservation with human chromosome Hsa15 (Fig. 5). As described in an aforementioned study, Tni13 matched Tni5 and Tni19 because chromosome Tni5 was derived from ancestral chromosome E, and chromosome Tni19 was derived from ancestral chromosome F by 3R. The other copy of chromosome E and chromosome F developed into chromosome Tni13 by fusion or fragmentation (*Jaillon et al., 2004*). Combining with neighborhood gene synteny result, it supports that two teleost *SMAD3* isoforms are generated from 3R.

## Teleost *smad3a* and *smad3b* genes differ in phylogenesis and gene structure

In order to confirm the relations and differences between teleost *smad3a* and *smad3b*, phylogenetic reconstruction and genomic structure analyses were performed. As shown in Fig. 3, the phylogenetic relationships were clearly displayed. Under the teleost clade, *smad3a* was clearly separated from *smad3b*, and spotted gar *SMAD3* occupied a separate clade. Thus, it could be speculated that *smad3a* and *smad3b* diverged from a common ancestral gene. According to the branch length, we could speculate that the evolution speed of teleost *smad3b* was faster than that of *smad3a*. Amphibians and reptiles occupy a special status in the evolution history: their genes may have suffered extreme selective pressure during this period, which might be reflected by the branch lengths of African clawed frog and anole lizard clades.

We also explored the conservation of gene structures of *smad3a* and *smad3b* in teleosts. Results showed that the genomic structures of *smad3a* and *smad3b* were highly conserved with minor differences: the fourth exon in *smad3a* was divided into two exons in *smad3b* by an additional intron and introns in *smad3a* were much longer than those in *smad3b*. The different exons were located in the linker region in *SMAD3* protein sequence which is supposed to regulate gene function. Introns that interrupt eukaryotic

protein-coding sequences are important indicators in eukaryotic evolution, and intron gain or loss rate has a significant correlation with the coding sequence evolution rate (*Carmel et al., 2007*). Genes can be regulated by various intronic properties, such as sequence, length, position and splicing, in the aspects of transcription initiation, transcription termination, genome organization and transcription regulation (*Chorev & Carmel, 2012*). Duplicate genes tend to diverge in regulatory and coding regions by amino acid-altering substitutions and/or alterations in exon-intron structure. Besides, the structural divergences have played a more important role in the evolution of duplicate genes than nonduplicate genes (*Xu et al., 2012*). Taken together, it could be inferred that the sequences of *smad3a* and *smad3b* had been changed under selection pressure and the different introns may regulate the expression and function of these two paralogs.

## Subfunctionalization of the Japanese flounder *smad3a* and *smad3b*

Gene duplication is believed to be the primary source of new genes and plays significant roles in the evolution of genomes and genetic systems (*Gu et al., 2003*; *Ohno, 1970*). The sequences and structures of gene pairs that originated from duplication will undergo rapid changes and have different evolutionary rates (*Zhang, Zhang & Rosenberg, 2002*). Generally, accumulation of detrimental mutations is probably the most common fate of one of the duplicates while the other copy maintains initial function (*Cañestro et al., 2013*). As described in the DDC model, degenerative mutations increased the probability of a duplicated gene's preservation, and the preservation of duplicated genes is related to the partitioning of ancestral functions but not to the evolution of new functions. Therefore, duplicated genes underwent three fates, namely, nonfunctionalization, subfunctionalization and neofunctionalization (*Force et al., 1999*). In our study, we hypothesized that *smad3a* and *smad3b* had undergone subfunctionalization in teleost after 3R according to plenty of analyses.

The gene structure of *smad3a* and *smad3b* differed in the lengths of introns and the fourth exon which was interrupted by an additional intron in *smad3b*. Beyond that, sequences of *smad3a* and *smad3b* shared high similarity. Motif scan results by MEME showed no significant differences (Fig. S4). These findings suggested that the two duplicates did not acquire novel functions.

As to the domains of *SMAD3*, the conserved exons between *smad3a* and *smad3b* were located in MH1 and MH2 domains, respectively, whereas the different exons were located in the linker region of *SMAD3*. MH1 and MH2 domains contributed to the DNA binding and protein association function of *SMAD3*, whereas linker region was rich of phosphorylation sites and acted as a negative region of *SMAD3* function (*Attisano & Lee-Hoeflich, 2001*). In accordance with the selection pressure analysis, both *SMAD3* paralogs were under purifying selection, and two significant mutated amino acid sites (219L, 221L) were predicted to have undergone strong positive selection among linker region in *smad3b* relative to *smad3a*. These positive selected sites may affect the function of linker region to regulate the interaction between MH1 and MH2 domains, resulting in the functional divergence between *smad3a* and *smad3b*.

The roles of *SMAD3*, such as in regulation of fibronectin, wound healing process, renal disease, and cell proliferation, were widely discussed in previous studies (*Isono et al., 2002*; *Schiller, Javelaud & Mauviel, 2004*; *Ten Dijke et al., 2002*; *Wang, Koka & Lan, 2005*). In the present study, we found that the expression patterns of *smad3a* and *smad3b* were different. Their expression levels were significantly different in brain, muscle, ovary, and spleen. The expression level of *smad3b* was higher than that of *smad3a* in brain, muscle, and ovary, contrary to those in spleen. Thus, a functional divergence exists between *smad3a* and *smad3b* in some biological processes in Japanese flounder. Further comprehensive in vitro and in vivo studies should be conducted to elucidate the functions of *smad3a* and *smad3b* in suitable model teleosts.

Overall, we conclude that the functions of teleost *smad3a* and *smad3b* shared conserved domain function, while functional divergence occurred because of their different evolution fates. These results provided sufficient evidence to conclude that the duplicated *SMAD3* genes had undergone subfunctionalization after 3R.

## CONCLUSION

In summary, we explored the origin of teleost *smad3a/3b* genes in this study and reported the general duplication of *SMAD3* resulting from 3R in teleosts. This study is the first to investigate the duplication of teleost *SMAD3* genes by selection pressure analysis. The results suggested probable subfunctionalization of duplicated teleost *SMAD3* genes. This study provided adequate information and new insights into the teleost *SMAD3* genes for further functional research in teleost.

### Funding

This work was supported by the National High-Tech Research and Development Program of China (2012AA10A402) and the National Natural Science Foundation of China (No. 31272646). The funders had no role in study design, data collection and analysis, decision to publish, or preparation of the manuscript.

### Grant Disclosures

The following grant information was disclosed by the authors:
National High-Tech Research and Development Program of China: 2012AA10A402.
National Natural Science Foundation of China: 31272646.

### Competing Interests

The authors declare that they have no competing interests.

### Author Contributions

- Xinxin Du performed the experiments, analyzed the data, wrote the paper, prepared figures and/or tables, reviewed drafts of the paper.
- Yuezhong Liu prepared figures and/or tables, reviewed drafts of the paper.
- Jinxiang Liu contributed reagents/materials/analysis tools, reviewed drafts of the paper.

- Quanqi Zhang contributed reagents/materials/analysis tools, reviewed drafts of the paper.
- Xubo Wang conceived and designed the experiments, reviewed drafts of the paper.

## Animal Ethics

The following information was supplied relating to ethical approvals (i.e., approving body and any reference numbers):

This research was conducted in accordance with the Institutional Animal Care and Use Committee of the Ocean University of China and the China Government Principles for the Utilization and Care of Vertebrate Animals Used in Testing, Research, and Training (State science and technology commission of the People's Republic of China for No. 2, October 31, 1988: http://www.gov.cn/gongbao/content/2011/content_1860757.htm).

## Data Deposition

The raw data has been supplied as Supplemental Dataset Files.

## Supplemental Information

Supplemental information for this article can be found online at http://dx.doi.org/10.7717/peerj.2500#supplemental-information.

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
