# Peer review of "Evolution history of duplicated smad3 genes in teleost: insights from Japanese flounder, Paralichthys olivaceus"

_PeerJ, doi:10.7717/peerj.2500_

## Round 0.1 · original submission · Major Revisions

· Academic Editor

Major Revisions

Please revise the manuscript according to the suggstions from the reviewers.

·

Basic reporting

The article has some language problems, that are likely due to English being not their native language. The text should be checked by (ideally) native speaker. I add some of these in the section "General Comments to the Authors". Also many of the references are out of date. For instance all reference to the impact of whole genome duplications are out of date and miss some of the more recent discussions on their role in speciation and neofunctionalization. Some claims are not supported by the experiments (may also be a language problem), in particular that a functional analysis was constructed. This is a purely descriptive report without functional analysis.
The figure legends need to be better described. It is not clear what Figure 4 shows.

Experimental design

The experiments are generally well done and convincing. One point is that the authors do not make it clear if the use the whole gene sequences for their phylogenetic analysis or use aligned sequnces (e.g. GBlocks) to account for different gene lengths.

Validity of the findings

The data is valid and adequately discussed.

Additional comments

Here are some, but certainly not all, phrases where the language needs to be adjusted.
what is a "vital vertebrate"?
significant roles in many aspects - not sure what is meant by this.
there is no "partial conclusion" (speculate, find evidence?)
genes do diverge not differentiate

·

Basic reporting

The English language used throughout the text is not clear enough. The objectives and methods used to achieve them are not well defined. Literature is well referenced and relevant. Figures are relevant with acceptable quality, well labelled and described.

Experimental design

I believe that the study is within scope of the journal but the research question is not well defined, although being relevant an meaningful. Rigorous investigations were performed but the results are not well presented and discussed in an appropriate way. Methods are not described with sufficient detail and sufficient information.

Validity of the findings

The findings of the study will not great impact and do not bring too much novelty to the scientific community. Data are robust and statistically sound but must be presented in appropriate way to make the manuscript more sound.

Additional comments

The manuscript by Du et al. investigate the duplicated smad3 genes and their eventual functional divergence in Teleost. The authors identified two smad3 paralogs in Japanese flounder. The synteny analyses and phylogenetic reconstruction revealed that the two smad3 isoforms (smad3a and smad3b) originated from a teleost-specific whole-genome duplication. The functional analysis further suggested that these two smad3 isoforms specific to teleost may have underwent sub-functionalyzation after duplication. Although I found the results interesting, the way that the study is presented and the text is written, make the manuscript unsound.

I would suggest organizing the manuscript as follow:
- Similarity search (Blast) and gene structure were performed to identify different smad3 isoforms,
-Synteny and phylogenetic reconstruction were conducted to confirm differences between smad3 isoforms and to infer their origination by whole-genome duplication,
- Functional analyses were done to study an eventual functional divergence of the identified isoforms.
More importantly, the authors have to explain in details why each type of analysis was conducted and how it was done. Please find below some examples that justify my comments.

1. In the abstract, the term “biological diversification” can mean many things. I suggest using lineage diversification because they refer to an evolutionary event that is specific to teleost lineage.
2. The term smad is not homogeneously used in the text. In some cases, it is SMAD or SMAD, in other it is written as smad. Please homogenise throughout the text.
3. In the introduction (lines 54-56), they authors indicated that smad isoforms in teleost are evidently originated from whole-genome duplication. They should clearly mention that this conclusion is from previous studies and consequently perform analyses in their study to verify if some of smad isoforms in teleost are not originated from fragment duplication or alternative splicing.
4. Introduction (lines 67-84), the authors should define or explain the differences between the terms non-, sub- and neo-functionalization.
5. Lines 100-102, the blast criteria must be indicated.
6. Line 106, the authors should indicate if they used the nucleotide or amino acid sequences. They should also mention clearly that they perform a phylogentic reconstruction including all smad isoforms from all vertebrates and a second phylogenetic reconstruction that include only smad3 isoforms.
7. Line 128, the authors should be more precise by indicating, for example, that they performed synteny comparisons of the fragment harbouring smads and their flanking genes.
8. Line 130, why only nine species were selected for this analysis? The criteria used to select these nine species should be indicated.
9. Line 202-203, the authors mentioned this: “smad3b is more likely to be the ancestral gene and smad3a derived from a duplication”. They further indicated (lines 295-297) that “ smad3b evolved faster than smad3a. I found these conclusions contradictory. If smad3b is the ancestral gene, it should be more similar to other vertebrate smad3 such as in spotted gar or elephant shark. Did the authors checked for that? If smad3b is the ancestral gene, it should me more similar to the variant found in the other vertebrates. The smad3a that is specific to teleost should be more different.
10. Lines 305-306, How different introns can regulate the expression and function? Was this previously demonstrated? This argument need to be explained or supported by references.
For all the raisons mentioned above, I cannot accept the publication of the manuscript its current form, although the results are interesting. I suggest major revisions including reorganizing and rewriting the text.

---

## Round 0.2 · Minor Revisions

· Academic Editor

Minor Revisions

Please revise the manuscript according to the remaining minor suggestions from the reviewer.

·

Basic reporting

The English language used throughout the text was edited in this revised version of the manuscript and I believe it is now clear enough. The objectives and methods used to achieve them are well defined. Literature is well referenced and relevant. Figures are relevant with acceptable quality, well labelled and described.

Experimental design

I believe that the study is within scope of the journal. The revised manuscript was reorganized and discussed in an appropriate way. Rigorous investigations were performed and the current results are well presented and discussed in an appropriate way. Methods of the current version are described with sufficient detail and sufficient information.

Validity of the findings

The findings of the study are interesting but do not bring too much novelty to the scientific community. Data are robust, statistically sound. They are presented in appropriate way in the revised version of the manuscript.

Additional comments

I really do not see the interest of using different abbreviations of gene name for the orthologous gene in human, house mouse and teleost without explaining that in the text. I suggest using the same abbreviation for human, house mouse and teleost orthologs and using different names for teleost paralogs. Just explain that there different abbreviations for the SMAD gene orthologs in NCBI but, for convenience reasons, in your study you used SMAD for all vertebrate orthologs and smad3a and smad3b for the variants identified in teleost. The way authors used gene names is very confusing and must be changed.

---

## Round 0.3 · accepted · Accept

· Academic Editor

Accept

This revised manuscript is accepted for publication in PeerJ.